# Beyond Top-K: Structured Sparsification for Compression in Pipeline Parallel

**Sameera Ramasinghe, Thalaiyasingam Ajanthan, Gil Avraham, Yan Zuo, & Alexander Long**
Pluralis Research
{sameera, aj,gil,yan,alexander}@pluralis.ai

## Abstract

In decentralized training, efficient communication is critical, particularly when training large-scale models over low-bandwidth, heterogeneous networks. Although gradient compression techniques have proven effective in Distributed Data-Parallel (DDP) settings, extending them to pipeline parallel (PP) training is challenging due to cumulative compression errors that exacerbate with network depth. In this work, we introduce a novel compression framework for PP that preserves the column space of activations and gradients instead of compressing individual elements. We derive tight theoretical error bounds and demonstrate the effectiveness of our method by training models over 80 Mbps connections, achieving up to 90% compression along with around $2\times$ training and $12\times$ inference throughput improvements.

## 1 Introduction

In decentralized training environments, particularly when training large-scale models over low-bandwidth and heterogeneous networks, efficient communication between nodes is critical. Traditionally, most research has concentrated on compressing gradients in Distributed Data-Parallel (DDP) settings, motivated by the observation that weight gradients are highly redundant and can be aggressively compressed—via techniques such as TopK sparsification Lin et al. (2017), low-rank projections Vogels et al. (2019), and quantization Seide et al. (2014)—without significantly affecting convergence. These methods have demonstrated impressive compression rates and have been widely adopted in DDP.

However, extending these compression techniques to pipeline parallel (PP) training presents unique challenges. Recent studies Bian et al. (2024); Song et al. (2023) have shown that sparsification strategies effective in DDP do not translate well to PP settings. In PP, both the forward and backward passes involve transmitting intermediate activations and gradients between layers rather than simply sharing gradients between model replicas. Consequently, each layer introduces its own compression error, and these errors can accumulate Song et al. (2023). We show that the compound error can potentially growing exponentially with network depth, which can severely degrade convergence.

Further, we argue that preserving the column space of activations and gradients is more critical in PP than retaining individual elements, as is the focus in DDP. Our analysis reveals that element-wise sparsification methods, such as TopK, disrupt the column space, leading to greater cumulative errors. Instead, we propose a column-wise sparsification strategy that better maintains the structural integrity of the information passed between layers, while achieving a better compression rate.

We derive tight upper bounds on the errors incurred under different compression schemes and provide theoretical insights into their behavior in PP settings. To validate our theoretical findings, we conduct experiments by training models in a decentralized environment characterized by low-bandwidth (80 Mbps) connections with stochastic variability. Our results demonstrate that the proposed method achieves up to a $90\%$ compression rate—substantially surpassing traditional TopK—and translates into around $2\times$ overall throughput gain during training and a $12\times$ gain during inference. Our findings suggest that rethinking compression strategies for PP—by focusing on preserving structural information rather than individual elements—can lead to substantial improvements in both training and inference efficiency.

This paper makes the following contributions:

- We identify and analyze the challenges associated with applying gradient compression techniques in pipeline parallel training, highlighting the role of error accumulation.
- We propose a novel compression framework that better preserves the column space of activations and gradients, improving convergence.
- We provide theoretical analysis with tight error bounds and validate our approach on low bandwidth, decentralized settings.

## 2 RELATED WORKS

**Model Parallelism and Data Parallelism.** Data parallelism (DP) replicates the model across multiple devices, with each worker processing different data batches and synchronizing gradients Li et al. (2014); Sergeev & Del Balso (2020). While effective, DP faces communication bottlenecks as models scale, prompting research into gradient compression methods like quantization Seide et al. (2014), sparsification Lin et al. (2017), and low-rank updates Vogels et al. (2019). Model parallelism (MP) instead partitions the model across devices, allowing training of large models beyond single-device memory limits Shoeybi et al. (2019). It is commonly implemented as tensor model parallelism (TP), which splits computations within layers, or pipeline parallelism (PP), which assigns different layers to different workers Narayanan et al. (2019). Both strategies introduce communication overhead, making compression a key optimization.

**Compression in Parallel Training.** Unlike DP, where gradient compression is widely studied, MP requires activation compression, which poses unique challenges since activations are not inherently low-rank Li et al. (2022); Bian et al. (2024). Recent approaches have explored autoencoder-based Hinton & Zemel (1993) and quantization-based Wang et al. (2022) compression, but excessive compression can degrade accuracy. Furthermore, existing system optimizations that improve DP efficiency may not translate to MP due to differences in communication patterns Agarwal et al. (2022). Understanding these trade-offs Song et al. (2023) is crucial for designing communication-efficient MP strategies that balance compression benefits with computational cost and accuracy retention.

## 3 ANALYSIS

### 3.1 COMPRESSION ERRORS IN PIPELINE PARALLEL TRAINING

A key distinction between DDP training and PP training lies in the nature of the information exchanged and the way compression is applied. In DDP, model weight gradients are exchanged after a number of training steps, allowing compression to be applied across the entire parameter gradient vector at once. Conversely, in PP training, activations and activation gradients must be exchanged between the layers of the model. This means that, unlike in DDP, each layer in PP training contributes independently to the overall compression error during each training iteration. As a result, the compression error is accumulated across layers, which poses unique challenges in maintaining model accuracy and stability. Given these differences, it is intriguing to analyze PP training under the specific constraints imposed by activation and acivation gradient compression. First, we show that the compound error in PP training can grow exponentially with the number of layers.

**Theorem 3.1.** *Consider a feedforward neural network with $L$ layers, where layer $l$ applies a (differentiable) function*

$$x_{l+1} = f_l(x_l), \quad l = 1, \ldots, L.$$

*Let $\nabla_L(x_l)$ denote the gradient of the final loss $\mathcal{L}$ with respect to the layer's input $x_l$. Suppose that: 1) The spectral norm of the Jacobian $\nabla f_l(x_l)$ is bounded above by $\nu > 0$ for all $l$, i.e. $\|\nabla f_l(x_l)\| \leq \nu$. 2) In backpropagation, an additional error $e_l$ is introduced at each layer $l$, with $\|e_l\| \leq e$ for some constant $e > 0$. Further, define $\varepsilon_l$ to be the* cumulative *error in the gradient at layer $l$. Then for $\nu > 1$, $\varepsilon_l$ can grow exponentially with the total number of layers $L$; in particular,*

$$\|\varepsilon_l\| \leq e \frac{\nu^{L-l+1} - 1}{\nu - 1},$$

*which is an exponential function of $L$ when $\nu > 1$.*

## 3.2 COLUMN-SPACE PRESERVATION IN PIPELINE PARALLELISM

We refer to the notation provided in Appendix 7.3 from here onward.

**Forward Pass.** Recall that the column space of the product $\mathbf{AB}$ of any two matrices $\mathbf{A}$ and $\mathbf{B}$ always lies in the column space of $\mathbf{A}$. Therefore, every subsequent linear transformation or attention mechanism operates within the column space of the activation matrix, and thus, preserving that space in compression is essential. Specifically, if $\mathbf{X}^l \in \mathbb{R}^{b \times n \times d}$ is replaced by some compressed $\widetilde{\mathbf{X}}^l$, then every projection $(\widetilde{\mathbf{X}}^l \mathbf{W}^l_{*,h})$ will remain within the subspace spanned by $\widetilde{\mathbf{X}}^l$. If the subspace of $\widetilde{\mathbf{X}}^l$ deviates significantly from that of $\mathbf{X}^l$, the transformed representations—queries, keys, values, and subsequent feed-forward inputs—will be misaligned.

**Backward Pass.** Similarly, on the backward pass, let $\nabla_l(\mathbf{X}^l_{\text{hidden}}) = \nabla_l(\mathbf{X}^{l+1})(\mathbf{W}^l_{p_2})^\top$. Any deviation in the column space of $\nabla_l(\mathbf{X}^l)$ will adversely affect the propagation of gradients, leading to inaccurate updates in earlier layers. Thus, maintaining fidelity in the column spaces becomes paramount to preserve accurate gradient flow.

### 3.2.1 A SUBSPACE-BASED METRIC FOR COMPRESSION QUALITY

To quantify the misalignment of the column space in a compressed activation (or gradient) matrix, consider a single instance $G \in \mathbb{R}^{n \times d}$ from a batch. Let $\widetilde{G}$ be its compressed version. Denote by $\mathbf{U}_r$ the top $r$ left singular vectors of $G$, and by $\widetilde{\mathbf{U}}_r$ the top $r$ left singular vectors of $\widetilde{G}$. We assess subspace preservation via the principal angles between these subspaces:

$$\left\| \sin \Theta(\mathbf{U}_r, \widetilde{\mathbf{U}}_r) \right\|, \tag{1}$$

where $\Theta(\mathbf{U}_r, \widetilde{\mathbf{U}}_r)$ is the diagonal matrix whose entries are the principal angles between the column spaces of $\mathbf{U}_r$ and $\widetilde{\mathbf{U}}_r$. Intuitively, this term captures the extent to which the compressed matrix $\widetilde{G}$ is "rotated" relative to $G$. If the principal angles are large, the subspaces are significantly misaligned, implying higher risk of distortion in both forward and backward passes. Consequently, $\| \sin \Theta(\mathbf{U}_r, \widetilde{\mathbf{U}}_r) \|$ serves as a concise, direct measure of the *error* introduced by compression.

By ensuring that compression schemes maintain small principal angles between the uncompressed and compressed subspaces, one can more effectively preserve the representational and gradient flow characteristics critical to stable training in pipeline-parallel architectures.

## 3.3 ON THE ERROR OF TOPK SPARSIFICATION

A common strategy for compressing gradients in DDP training is Top-K sparsification. In this method, only the top k% elements (by absolute value) of the gradient tensor are retained while the remaining entries are set to zero. When combined with error correction mechanisms, Top-K sparsification has been highly effective in DDP, achieving compression rates as high as 99% without degrading convergence.

However, in PP training, Top-K sparsification is considerably less effective. Recent studies Bian et al. (2024); Song et al. (2023) have shown that aggressive Top-K compression can significantly degrade convergence in PP. In the following, we derive a rigorous upper bound on the error induced by Top-K sparsification on the column space, confirming that this error can increase rapidly at high compression rates.

**Theorem 3.2.** *Let $G \in \mathbb{R}^{n \times d}$ be a random matrix whose entries $\{G_{ij}\}_{1 \leq i \leq n,\, 1 \leq j \leq d}$ are i.i.d. sub-Gaussian with mean zero and variance proxy $\sigma^2$. Assume $G$ has rank $r$, and that its $r$th singular value satisfies*

$$\sigma_r(G) \geq \beta > 0.$$

*For any fraction $0 < x < 1$, define $\widetilde{G}$ by* masking *(setting to zero) all entries of $G$ whose absolute value does* not *exceed the $(x\,n\,d)$th order statistic:*

$$|G_{(1)}| \leq |G_{(2)}| \leq \cdots \leq |G_{(nd)}|, \quad t_x := |G_{(\lceil x\,n\,d \rceil)}|.$$

*Then set*

$$\widetilde{G}_{ij} = \begin{cases} 0, & \text{if } |G_{ij}| \leq t_x, \\ G_{ij}, & \text{if } |G_{ij}| > t_x. \end{cases}$$

*With high probability (in $n, d$), the Frobenius norm of the sine of the principal angles between the top-$r$ left singular vectors of $G$ (denoted $U_r$) and those of $\widetilde{G}$ (denoted $\widetilde{U}_r$) is bounded by*

$$\| \sin \Theta(\widetilde{U}_r, U_r) \|_F \leq C \sigma \sqrt{\frac{r\,x\,n\,d}{\beta^2}} \sqrt{\log\left(\frac{2}{1-x}\right)},$$

*where $C > 0$ is a universal constant (independent of $n, d, x, \sigma, \beta$).*

**Discussion.** The error bound in Theorem 3.2 $C \sigma \sqrt{\frac{r\,x\,n\,d}{\beta^2}} \sqrt{\log\left(\frac{2}{1-x}\right)}$ captures several important factors influencing the accuracy of the masked matrix approximation. The dependence on $\sqrt{r}$ suggests that a lower-rank approximation of $G$ leads to a smaller error, meaning that if $G$ is well-approximated by a low-rank structure, the impact of the masking is less severe. Furthermore, the parameter $\beta$ reflects the minimum separation among the nonzero singular values of $G$; a larger $\beta$ indicates a clearer distinction between significant and insignificant singular values, which in turn reduces the error bound.

A critical aspect of the bound is the term $\sqrt{x \log\left(\frac{1}{1-x}\right)}$, which represents the effect of the fraction $x$ of masked entries. As more aggressive compression is applied—meaning that $x$ approaches 1 and almost all entries are masked—the logarithmic term $\log\left(\frac{2}{1-x}\right)$ diverges. In the limit as $x \to 1$, this divergence causes the overall bound to grow without bound, indicating that the error in approximating the subspace becomes arbitrarily large. Such an unbounded increase in error under extreme compression is particularly concerning in PP settings, where errors introduced at each layer can accumulate and significantly degrade convergence.

## 4    IMPROVING THE ERROR BOUND WITH COLUMN-SPARSIFICATION

In contrast to element-wise compression methods, such as Top-K sparsification which can induce to severe errors in the resultant column space, we show that leveraging *column-sparsification* to better preserves the structural properties of the data. Instead of masking individual entries of the gradient or activation matrices, we mask entire columns—specifically, those with the smallest $\ell_2$ norms. This strategy aims to maintain the integrity of the column space, which is critical for preserving the overall subspace structure during the training process.

The following theorem formalizes the error bound when column-sparsification is applied.

**Theorem 4.1.** *Let $G \in \mathbb{R}^{n \times d}$ be a matrix of rank $r$ whose nonzero singular values are all at least $\beta > 0$. Consider masking the $x\%$ columns of $G$ having the smallest $\ell_2$ norm, resulting in a matrix $\hat{G}$. Denote by $\mathbf{U}_r$ and $\tilde{\mathbf{U}}_r$ the matrices containing the top $r$ left singular vectors of $G$ and $\hat{G}$, respectively. Then,*

$$\| \sin \theta(\tilde{\mathbf{U}}_r, \mathbf{U}_r) \|_F \leq \frac{\sqrt{2rx\,d} \, \max_{i \in \mathcal{I}} \|G_{:,i}\|_2}{\beta},$$

*where $\mathcal{I}$ denotes the (masked) column indices.*

**Discussion.** This theorem shows that by masking columns with small $\ell_2$ norms, the deviation between the subspaces spanned by the top $r$ singular vectors of $G$ and $\hat{G}$ is controlled by the term $\sqrt{x\,d}$ multiplied by the maximum norm of the masked columns, normalized by the spectral lower bound $\beta$. In other words, if the columns being discarded are indeed of small magnitude (i.e., $\max_{i \in \mathcal{I}} \|G_{:,i}\|$ is small), then the subspace perturbation is guaranteed to be small, even if a non-negligible fraction $x$ of the columns is masked.

This result contrasts sharply with the behavior observed in element-wise Top-K sparsification, where the error bound may grow rapidly—indeed, diverging in the limit of aggressive compression. Here,

the column-sparsification method maintains a more stable error behavior: the error scales as $\sqrt{x\,d}$, which grows more gracefully with the compression rate. Consequently, this approach mitigates the cumulative error problem inherent in PP training, where repeated compression across layers could otherwise lead to significant degradation in convergence.

Thus, column-sparsification offers a promising alternative for reducing communication overhead while preserving essential structural information, enhancing both the theoretical and practical performance of large-scale decentralized training systems.

# 5 EXPERIMENTS

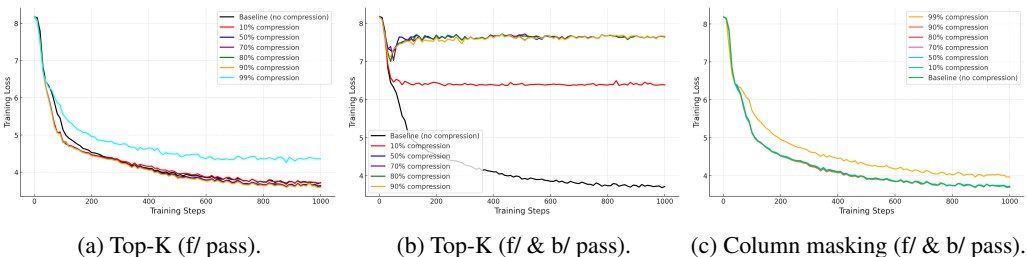

(a) Top-K (f/ pass).  (b) Top-K (f/ & b/ pass).  (c) Column masking (f/ & b/ pass).

Figure 1: **Comparison with Top-K:** (a) Compressing only the forward pass (activations) with Top-K allows higher compression rates. (b) Compressing both the forward and backward passes (activations and gradients) with Top-K leads to severe performance degradation. (c) Our proposed column-masking approach enables compressing both forward and backward passes with higher compression rates while maintaining performance.

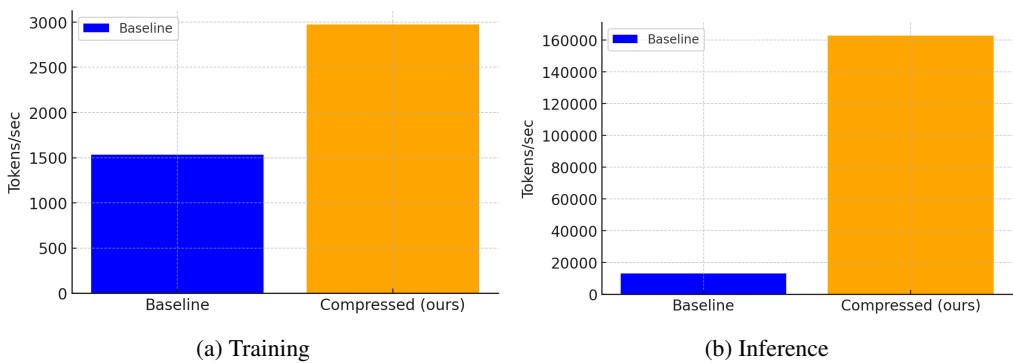

(a) Training  (b) Inference

Figure 2: **Throughput gain with** $90\%$ **compression**. We achieve around $2\times$ and $12\times$ gains over the non-compressed model.

## 5.1 EXPERIMENTAL SETUP

We evaluate our method on language-modeling tasks using decoder-only architectures Brown et al. (2020) trained on the C4 dataset Raffel et al. (2019). The model configurations are based on the Llama 3 architecture Dubey et al. (2024). Specifically, we employ a context length of 1024, an embedding dimension of 512, 24 attention heads, and 8 layers. For optimization, we use a base learning rate $\eta = 3 \times 10^{-4}$ with a warm-up phase followed by linear decay, a weight decay of 0.01, and a batch size of 32.

We adopt GPipe Huang et al. (2019) for pipeline parallelism (via `torch.distributed.pipelining`) and integrate our compression method directly into its dataflow. The model is partitioned across 8 A10g GPUs, assigning one layer per GPU. To simulate heterogeneous network conditions, We designate the connections between the 2nd–3rd, 4th–5th, and 6th–7th GPUs as low-bandwidth links, with bandwidth sampled from $\mathcal{N}(80, 16)$ (Mb/s),

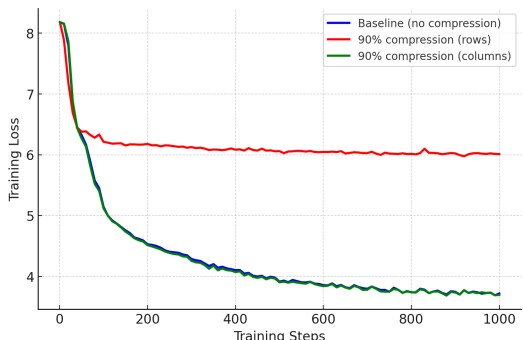

Figure 3: **Comparison against row-masking.** Validating our argument that the preservation of the column space is more critical, the model convergence is disrupted severely when the rows are masked.

while all other links operate at higher bandwidth sampled from $\mathcal{N}(1024, 200)$. Note that in PP, the lowest link bottlnecks the training.

## 5.2 COMPARISON WITH TOP-K

We compare the performance of our proposed method against top-K under varying compression rates. Fig. 3 presents the results. As shown in Fig. 3, even with top-K, compressing only the forward pass maintains good convergence, even at aggressive compression rates. However, when backward gradients are also compressed, training diverges significantly—even at mild compression rates (e.g., 10%). This highlights the greater sensitivity of gradients to compression errors compared to activations. In contrast, our column-based compression preserves convergence and matches the performance of the uncompressed baseline, even at a 90% compression rate. Note that this aligns with our theoretical predictions in Theorem 3.2 and 4.1.

## 5.3 THROUGHPUT GAINS

Achieving a 90% compression rate leads to significant throughput improvements over the baseline. Fig. 2b presents the results, showing that our method achieves approximately a 2× increase in training throughput and a 20× boost in inference throughput. These gains can substantially reduce latency in decentralized settings, making large-scale training and deployment more efficient.

## 5.4 COMPARISON WITH ROW MASKING

Our compression method is based on the premise that preserving the column space of activations and gradients is critical. This is already supported by our results in Fig. 2. To further validate this, we apply masking over rows instead of columns. Fig. 3 presents the results, showing that row masking leads to significant performance degradation, reinforcing the importance of column-space preservation.

## 6 CONCLUSION

We introduced a novel compression framework for pipeline parallel training that preserves the column space of activations and gradients, mitigating cumulative compression errors. Our approach achieves up to 90% compression, leading to a 2× training throughput and 12× inference speedup over low-bandwidth networks. Theoretical analysis and large-scale experiments validate its effectiveness, highlighting the importance of structural preservation in PP compression.

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

## 7 APPENDIX

### 7.1 THEORETICAL RESULTS

#### 7.1.1 PROOF FOR THEOREM 3.1

*Proof.* Recall the usual chain rule for the gradient of the final loss $\mathcal{L}$ with respect to the input $x_l$ of layer $l$:

$$\nabla_L(x_l) = \nabla_L(x_{l+1}) \nabla f_l(x_l).$$

Assume that at each layer we introduce an error in the gradient. Let

$$\varepsilon_l = (\text{true gradient at layer } l) - (\text{observed/propagated gradient at layer } l).$$

When moving from layer $l$ to layer $l - 1$, the error recursion becomes:

$$\varepsilon_{l-1} = \varepsilon_l \nabla f_{l-1}(x_{l-1}) + e_{l-1},$$

where $e_{l-1}$ is the *newly introduced* error at layer $l - 1$. Unfolding this backwards gives a general expansion:

$$\varepsilon_l = e_l + \sum_{j=l+1}^{L} \left( \prod_{i=l}^{j-1} \nabla f_i(x_i) \right) e_j.$$

Taking the norm and using the assumption $\|\nabla f_i(x_i)\| \leq \nu$ and $\|e_j\| \leq e$, we get:

$$\|\varepsilon_l\| \leq \sum_{j=l}^{L} \left( \prod_{i=l}^{j-1} \|\nabla f_i(x_i)\| \right) \| e_j \| \leq \sum_{j=l}^{L} \nu^{j-l} e = e \sum_{k=0}^{L-l} \nu^k,$$

where $k = j - l$. This geometric sum is

$$\sum_{k=0}^{L-l} \nu^k = \frac{\nu^{L-l+1} - 1}{\nu - 1} \quad (\text{valid for } \nu \neq 1).$$

Hence,

$$\|\varepsilon_l\| \leq e \frac{\nu^{L-l+1} - 1}{\nu - 1}.$$

Since $\nu > 1$, $\nu^{L-l+1}$ grows exponentially in $L$. $\qquad\square$

#### 7.1.2 PROOF FOR THEOREM 3.2

*Proof.* By Wedin's Theorem Stewart (1990), we get

$$\| \sin \theta(\mathbf{U}_r, \tilde{\mathbf{U}}_r)\|_F^2 + \| \sin \theta(\mathbf{V}_r, \tilde{\mathbf{V}}_r)\|_F^2 \leq \frac{\|\mathbf{U}_r^T \Delta\|_F^2 + \|\Delta \mathbf{V}_r\|_F^2}{\delta^2} \tag{2}$$

where

$$\delta = \min\{ \min_{1 \leq i \leq r, r+1 \leq j \leq n} |\sigma_i - \tilde{\sigma}_j|, \min_{i \leq i \leq r} \sigma_i \} > 0.$$

, $\Delta = G - \hat{G}$, and $\mathbf{V}_r, \tilde{\mathbf{V}}_r$ are the right singular vectors of the rank-r SVD of $G$ and $\hat{G}$, respectively. Then, we get,

$$\| \sin \theta(\tilde{\mathbf{U}}_r, \mathbf{U}_r)\|_F^2 \leq \frac{2r \|\Delta\|_F^2}{\delta^2}, \tag{3}$$

since $\|\mathbf{U}_r \Delta\|_F^2 \leq \|\mathbf{U}_r\|_F^2 \|\Delta\|_F^2$, $\|\mathbf{V}_r \Delta\|_F^2 \leq \|\mathbf{V}_r\|_F^2 \|\Delta\|_F^2$, and $\|\mathbf{U}_r\|_F^2, \|\mathbf{V}_r\|_F^2 = \|\mathbf{I}_r\|_F^2 = r$.

Note that,

$$\|\Delta\|_F^2 = \sum_{\substack{i,j \\ |G_{ij}| \leq t_x}} \Delta_{ij}^2 \leq \sum_{\substack{i,j \\ |G_{ij}| \leq t_x}} t_x^2 = \left| \{(i,j) : |G_{ij}| \leq t_x\} \right| t_x^2.$$

By definition of $t_x$, it is the $(x\,n\,d)$-th smallest absolute value. Thus exactly $x\,n\,d$ of the entries (up to rounding) satisfy $|G_{ij}| \le t_x$. Hence

$$\big|\{(i,j)\colon |G_{ij}| \le t_x\}\big| \;\le\; x\,n\,d.$$

Therefore,

$$\|\Delta\|_F^2 \;\le\; (x\,n\,d)\,t_x^2, \quad\text{and}\quad \|\Delta\|_F \;\le\; \sqrt{x\,n\,d}\;t_x.$$

Consequently,

$$\|G - \widetilde{G}\| \;=\; \|\Delta\| \;\le\; \|\Delta\|_F \;\le\; \sqrt{x\,n\,d}\;t_x.$$

Since $G_{ij}$ are i.i.d. sub-Gaussian$(\sigma^2)$, the random variable $|G_{ij}|$ has a sub-exponential tail with parameter proportional to $\sigma$. Thus:

$$\mathbb{P}(|G_{i,j}| > t) \le 2\exp\Big(-\frac{ct^2}{\sigma^2}\Big). \tag{4}$$

Inverting this for the event $G_{i,j} \le t$ at probability $x$ gives,

$$x = \mathbb{P}(|G_{i,j}| < t) = 1 - \mathbb{P}(|G_{i,j}| > t) \ge 1 - 2\exp\Big(-\frac{ct^2}{\sigma^2}\Big) \tag{5}$$

with high probability. Then,

$$1 - x \le 2\exp\Big(-\frac{ct^2}{\sigma^2}\Big) \tag{6}$$

$$\exp\Big(\frac{ct^2}{\sigma^2}\Big) \le \frac{2}{1-x} \tag{7}$$

$$t_x \le \frac{\sigma\sqrt{\log\Big(\frac{2}{1-x}\Big)}}{c} \tag{8}$$

$$\|\Delta\|_F \;\le\; \sigma\sqrt{x\,n\,d}\,\frac{\sqrt{\log\Big(\frac{2}{1-x}\Big)}}{c}. \tag{9}$$

$$\|\sin\Theta(\widetilde{U}_r, U_r)\|_F \;\le\; C\,\sigma\,\sqrt{\frac{r\,x\,n\,d}{\beta^2}}\,\sqrt{\log\Big(\frac{2}{1-x}\Big)}, \tag{10}$$

with high probability. $\qquad\square$

### 7.1.3   PROOF FOR THEOREM 4.1

*Proof.* Let $\Delta = G - \hat{G}$. By construction, $\Delta_{:,i} = 0$ for unmasked columns and $\Delta_{:,i} = G_{:,i}$ for columns $i \in \mathcal{I}$. Hence all columns of $\Delta$ are exactly the $x\%$ fraction of $G$'s columns with the smallest norms. In particular,

$$\|\Delta\|_F^2 \;=\; \sum_{i\in\mathcal{I}} \|G_{:,i}\|_2^2 \;\le\; |\mathcal{I}|\,\max_{i\in\mathcal{I}}\|G_{:,i}\|_2^2.$$

Since $\mathcal{I}$ is of size $x\%$ of $d$ columns, we get $|\mathcal{I}| = x\,d$ (assuming for simplicity that $x\,d$ is an integer), thus

$$\|\Delta\|_F^2 \;\le\; x\,d\,\max_{i\in\mathcal{I}}\|G_{:,i}\|_2^2.$$

Next, we use a standard subspace perturbation bound (a variant of Wedin's theorem or Davis–Kahan). In one of its common forms, for matrices $G$ and $\hat{G}$ both of rank at least $r$,

$$\|\sin\theta(\tilde{\mathbf{U}}_r, \mathbf{U}_r)\|_F \ \leq \ \frac{\sqrt{2r}\,\|\Delta\|_2}{\sigma_r(G) - \sigma_{r+1}(G)},$$

when $\sigma_r(G)$ is well separated from $\sigma_{r+1}(G)$. In the simpler scenario where $G$ has rank $\geq r$ and $\sigma_r(G) \geq \beta > 0$, we can take $\sigma_{r+1}(G) = 0$ (or $\beta$ is the gap). Then

$$\|\sin\theta(\tilde{\mathbf{U}}_r, \mathbf{U}_r)\|_F \ \leq \ \frac{\sqrt{2}\,\|\Delta\|_2}{\beta}\,\sqrt{r} \ \leq \ \frac{\sqrt{2r}\,\|\Delta\|_F}{\beta},$$

since $\|\Delta\|_2 \leq \|\Delta\|_F$.

we can proceed by simply noting:

$$\|\Delta\|_F \ \leq \ \sqrt{x\,d}\,\max_{i\in\mathcal{I}}\|G_{:,i}\|_2.$$

Then,

$$\|\sin\theta(\tilde{\mathbf{U}}_r, \mathbf{U}_r)\|_F \leq \frac{\sqrt{2r}\|\Delta\|_F}{\beta} \ \leq \ \frac{\sqrt{2rx\,d}\,\max_{i\in\mathcal{I}}\|G_{:,i}\|_2}{\beta},$$

□

## 7.2 Tightness Analysis of the Perturbation Bound

In our analysis, we utilize a version of Wedin's *sin* theorem which provides the following bound on the perturbation of the singular subspaces:

$$\|\sin\Theta(U_r, \tilde{U}_r)\|_F^2 \ + \ \|\sin\Theta(V_r, \tilde{V}_r)\|_F^2 \ \leq \ \frac{\|U_r^T\Delta\|_F^2 \ + \ \|\Delta V_r\|_F^2}{\delta^2}, \tag{11}$$

where $\Delta = G - \hat{G}$ is the perturbation, and

$$\delta \ = \ \min\Big\{ \min_{\substack{1\leq i\leq r \\ r+1\leq j\leq n}} \big|\sigma_i \ - \ \tilde{\sigma}_j\big|,\ \min_{1\leq i\leq r}\sigma_i\Big\} \ > \ 0.$$

Above, $U_r$ (resp. $V_r$) and $\tilde{U}_r$ (resp. $\tilde{V}_r$) are the left (resp. right) singular vectors of $G$ and $\hat{G}$ corresponding to the top $r$ singular values, and $\sigma_i$ ($\tilde{\sigma}_j$) denotes the singular values of $G$ ($\hat{G}$).

A standard approach to further bound the right-hand side of equation 11 is to invoke a norm inequality for products of matrices. Specifically, for the term involving $U_r$, we can write:

$$\|U_r^T\Delta\|_F^2 \ \leq \ \|U_r^T\|_F^2\,\|\Delta\|_F^2. \tag{12}$$

Since $U_r$ is an $m \times r$ matrix with orthonormal columns, its Frobenius norm satisfies

$$\|U_r\|_F^2 \ = \ \sum_{j=1}^{r}\big\|(U_r)_j\big\|_2^2 \ = \ r.$$

Thus, equation 12 becomes

$$\|U_r^T\Delta\|_F^2 \ \leq \ r\,\|\Delta\|_F^2.$$

A similar bound holds for the term $\|\Delta V_r\|_F^2$, because $V_r$ is also an orthonormal basis (this time of size $d \times r$), yielding

$$\|\Delta V_r\|_F^2 \ \leq \ r\,\|\Delta\|_F^2.$$

Combining these estimates in equation 11 gives

$$\|\sin\Theta(U_r, \tilde{U}_r)\|_F^2 \ + \ \|\sin\Theta(V_r, \tilde{V}_r)\|_F^2 \ \leq \ \frac{2r\,\|\Delta\|_F^2}{\delta^2}.$$

It is important to note that this inequality is *sharp* in the worst-case sense. Indeed, the inequality

$$\|U_r\Delta\|_F^2 \ \leq \ \|U_r\|_F^2\,\|\Delta\|_F^2$$

is a direct consequence of the submultiplicative property of the Frobenius norm when $U_r$ has orthonormal columns. Furthermore, it becomes an equality when the columns of $\Delta$ lie entirely in the column space spanned by $U_r$. In other words, if $\Delta$ is chosen such that $\text{range}(\Delta) \subseteq \mathcal{R}(U_r)$, then

$$\|U_r \Delta\|_F^2 \;=\; \|U_r\|_F^2 \, \|\Delta\|_F^2 \;=\; r \, \|\Delta\|_F^2.$$

Moreover, one can construct perturbations where the entire contribution in the bound of equation 11 is accounted for by *only one* of the subspaces. For instance, it is possible to have

$$\|\sin \Theta(V_r, \tilde{V}_r)\|_F^2 \;=\; 0$$

while

$$\|\sin \Theta(U_r, \tilde{U}_r)\|_F^2 \;\approx\; \frac{2r \, \|\Delta\|_F^2}{\delta^2}.$$

This scenario demonstrates that the bound is tight in the sense that, even if the perturbation affects only one of the two subspaces ($U_r$ or $V_r$), the dependence on $r$, $\|\Delta\|_F$, and $\delta$ cannot be significantly improved without additional assumptions (e.g. spectral gaps, specific structures in $\Delta$, etc.).

In summary, the derived bound

$$\|\sin \Theta(U_r, \tilde{U}_r)\|_F^2 \;\leq\; \frac{2r \, \|\Delta\|_F^2}{\delta^2},$$

as well as the intermediate inequality

$$\|U_r \Delta\|_F^2 \;\leq\; \|U_r\|_F^2 \, \|\Delta\|_F^2,$$

are indeed tight in a worst-case scenario. This tightness implies that our perturbation analysis accurately captures the potential sensitivity of the singular subspaces to adversarial or maximally aligned perturbations.

## 7.3 TRANSFORMER BLOCK

In this section, we briefly review the structure of a transformer block before introducing our compression approach. Let $\mathbf{X}^l \in \mathbb{R}^{b \times n \times d}$ represent the input to the $l^{\text{th}}$ layer, where $b$ is the batch size, $n$ the sequence length, and $d$ the embedding dimension. For each attention head $h$ (with $h = 1, \ldots, H$ and $d_H = d/H$), the input is projected into query, key, and value spaces via linear transformations:

$$\mathbf{X}_{Q,h}^l = \mathbf{X}^l \, \mathbf{W}_{Q,h}^l, \quad \mathbf{X}_{K,h}^l = \mathbf{X}^l \, \mathbf{W}_{K,h}^l, \quad \mathbf{X}_{V,h}^l = \mathbf{X}^l \, \mathbf{W}_{V,h}^l,$$

where $\mathbf{W}_{Q,h}^l, \mathbf{W}_{K,h}^l, \mathbf{W}_{V,h}^l \in \mathbb{R}^{d \times d_H}$. Since the product of any matrix with a weight matrix lies in the column space of the original matrix, each of $\mathbf{X}_{Q,h}^l$, $\mathbf{X}_{K,h}^l$, and $\mathbf{X}_{V,h}^l$ resides in the column space of $\mathbf{X}^l$.

For each head, the attention output is computed by applying the softmax function to the scaled dot-product of queries and keys, and then multiplying by the values:

$$\mathbf{X}_{\text{head},h}^l = \text{softmax}\left( \frac{\mathbf{X}_{Q,h}^l (\mathbf{X}_{K,h}^l)^\top}{\sqrt{d_H}} \right) \mathbf{X}_{V,h}^l.$$

The outputs from all $H$ heads are concatenated and fed through a feed-forward network with a residual connection:

$$\mathbf{X}_{\text{concat}}^l = \text{Concat}\Big( \mathbf{X}_{\text{head},1}^l, \, \mathbf{X}_{\text{head},2}^l, \, \ldots, \, \mathbf{X}_{\text{head},H}^l \Big),$$

$$\mathbf{X}_{\text{attn}}^l = \mathbf{X}_{\text{concat}}^l \, \mathbf{W}_{p_1}^l + \mathbf{X}^l,$$

$$\mathbf{X}_{\text{hidden}}^l = f_{\text{relu}}\Big( \mathbf{X}_{\text{attn}}^l \, \mathbf{W}_1^l \Big),$$

$$\mathbf{X}^{l+1} = \mathbf{X}_{\text{hidden}}^l \, \mathbf{W}_{p_2}^l + \mathbf{X}_{\text{attn}}^l,$$

with weight matrices $\mathbf{W}_{p_1}^l \in \mathbb{R}^{d \times d}$, $\mathbf{W}_1^l \in \mathbb{R}^{d \times d_{\text{ff}}}$, and $\mathbf{W}_{p_2}^l \in \mathbb{R}^{d_{\text{ff}} \times d}$. For simplicity, we have omitted layer normalization steps, which do not affect our derivations. Typically, the feed-forward dimension $d_{\text{ff}}$ is chosen to be an integer multiple of $d$.

