# OpenReview forum: "Beyond Top-K: Structured Sparsification for Compression in Pipeline Parallel"
_ICLR.cc/2025/Workshop/MCDC — MCDC @ ICLR 2025_

### Official Review · Reviewer_ebQK · 2025-02-28

**Rating:** 8
**Confidence:** 4
**Fit:** 4

**Summary:**

This paper proposes a communication compression technique that works well with pipeline parallelism-based decentralized training for large models. The authors first demonstrate how and why top-k compression fails to do well in a pipeline parallelism setting. Then, a detailed analysis shows the benefits of preserving the column space using column-wise magnitude-based compression. The results show that it is possible to achieve up to 90% compression without any loss in performance, with 2x training and 12x inference throughput benefits.

**Reason For Giving A Higher Score:**

The paper is well written and clarifies the problem with existing compression techniques quite well. I believe the contributions are unique and can provide interesting insights into designing compression techniques for pipeline parallelism-based training mechanisms.

**Reason For Giving A Lower Score:**

NA

**Strengths And Weaknesses:**

Strengths:

1. The paper is very well-written and motivates the contributions well.

2. The detailed analysis of how and why top-k fails is intriguing and necessary to understand the need for better compression strategies for pipeline parallelism.

Weaknesses:

1. More details about the experimental setup can help to understand the nuances. For example, was error compensation used to fix the performance loss by top-k? How many rounds of communication does it take for models to converge with the new compression technique?

**Suggestions:**

Please provide additional experimental details as mentioned above, as well as compare with quantization based compression techniques as well.

---

### Official Review · Reviewer_eVPC · 2025-03-05

**Rating:** 7
**Confidence:** 4
**Fit:** 3

**Summary:**

The authors note that while DP uses top-k compression, it's harder to use for PP because the error compounds as the number of layers increases.
Instead, they propose to compress by pruning entire column based on the L2 norm.

**Reason For Giving A Higher Score:**

Simple method that shows clear empirical gain.

**Reason For Giving A Lower Score:**

More studies, across different model scales and number of PP stages, would be interesting.

**Strengths And Weaknesses:**

# Strengths

* The proposed method is extremely simple to implement, and can have a great impact
 * Empirical results vs top-k (particularly when pruning both the forward and backward pass) are significant
* The reasoning behind their method is sound

# Weaknesses

It is noted that "the compound error in PP training can grow exponentially with the number of layers". However, it'd make sense to only do compression between pipeline *stage* rather than between every layers. I'd be curious to understand how well the proposed compression would fare then vs top-k. In particular, we could do a 4-stages, and thus only compression the first 1/4th layer instead of the first few layers -- which is much more harmful! In that more realistic case, how badly would top-k fare vs the proposed method?

**Suggestions:**

* how many parameters are in the model?
* how well top-k vs l2-column-pruning fare as we scale the model?

---

### Official Review · Reviewer_Ui2Z · 2025-03-05

**Rating:** 7
**Confidence:** 3
**Fit:** 4

**Summary:**

This paper proposed a compression method for Pipeline Parallel that drops columns with low norms rather than elements. This method helps preserve the column space of the compressed matrix. The authors showed theoretically and empirically how this method improves upon standard element top-k compression.

**Reason For Giving A Higher Score:**

* The paper validates the method with both proof and experiment. For example, the advantage of selecting columns is shown with both proof and experiment comparison with the row-masking variant.

**Reason For Giving A Lower Score:**

* It will be better to compare with other compression methods mentioned in related works, e.g., low-rank compression and quantization.

**Strengths And Weaknesses:**

Strength:
* The paper is well-motivated. The authors explained how errors can pile up in PP, suggesting a better compression method is necessary for DP, and provided theoretical proof of why selecting columns is helpful.
* The result of f/ & b/ pass is very significant. Using column masking made training possible where top-k breaks.


Weakness:
* The paper only performs experiments on one model and one dataset. Testing multiple models + datasets or different model scales can better assess the effectiveness of the proposed method.

**Suggestions:**

* Most of the proof connects compression rate with misalignment of the column space. However, I still don't understand how misalignment connects to error in Theorem 3.1 and training loss decrease in the experiments. Is it omitted because it is too obvious?
* Figure 2 didn't show too much information for the space it occupies. It may be better to use a 2x2 table instead. But, if there is result throughput for other compression ratios, adding them to the figure will be very nice.
* It is unclear whether structured sparsification is helpful only for PP or generally suitable for pretrained language models. If it is applied to the gradient in DDP, is it better than Top-k?
* In Figure 1, it will be better to use a consistent color scheme for different compression rates, e.g. baseline is black, 10% is red, and then put top-k and column masking in one figure, shown with different line style, e.g. - - line for column masking, and ... line for top-k so that it will be easier to compare the two methods.

---

### Decision · Program_Chairs · 2025-03-06

**Decision:**

Accept

**Comment:**

This paper has been appreciated by all reviewers. We recommend taking suggestions into consideration for the final version of the manuscript.